# Research on Spatiotemporal Land Deformation (2012–2018) over Xi'an, China, with Multi-Sensor SAR Datasets

**Mimi Peng [1], Chaoying Zhao [1,2,*], Qin Zhang [1,2], Zhong Lu [3] and Zhongsheng Li [1]**

1   School of Geology Engineering and Geomatics, Chang'an University, Xi'an 710054, China; 2018026013@chd.edu.cn (M.P.); zhangqinle@263.net.cn (Q.Z.); Lizhsh@chd.edu.cn (Z.L.)
2   State Key Laboratory of Geo-Information Engineering, Xi'an 710054, China
3   Roy M. Huffington Department of Earth Sciences, Southern Methodist University, Dallas, TX 75275, USA; zhonglu@mail.smu.edu
*   Correspondence: zhaochaoying@163.com; Tel.: +86-29-82339251

**Abstract:** The ancient city of Xi'an, China, has been suffering severe land subsidence and ground fissure hazards since the 1960s, mainly due to the over-withdrawal of groundwater and large-scale urban construction. This has threatened and will continue to threaten the stability of urban infrastructure, such as the construction and operation of high buildings and subway lines. It is necessary to map the spatiotemporal variations of land subsidence over Xi'an, and to analyze their causes and the correlation with underground water level changes and ground fissure deformation. Time series of land subsidence were observed with the interferometric synthetic aperture radar (InSAR) technique, using multi-sensor SAR datasets from 2012 to 2018. Four land subsidence rate maps over Xi'an city were retrieved from TerraSAR-X, ALOS/PALSAR2, and Sentinel-1 data, each with different tracks. The InSAR derived results were then cross-validated with three independent SAR data stacks, and calibrated with GPS and leveling observations. Next, the spatiotemporal evolutions of three main regional land subsidence zones were quantitatively analyzed in detail, and the surface deformation of the Xi'an subway network was spatially analyzed. Third, the correlations between land subsidence and ground water withdrawal, ground fissure deformation, landforms, and faults were intensively analyzed. Finally, a flat lying sill model with distributed contractions was implemented to model the InSAR deformation over one typical subsidence zone, which further suggested that the ground deformation was mainly caused by groundwater withdrawal. This systematic research can provide sound evidence to serve decision-making for land subsidence mitigation in Xi'an, and may also guide land subsidence research in other cities.

**Keywords:** Xi'an; land subsidence; ground fissures; multi-sensor SAR datasets; subway lines

## 1. Introduction

Land subsidence caused by the compaction of aquifer systems is a worldwide engineering and geological problem in urban areas, which are heavily dependent on groundwater supplies. The Wei River Basin is one of the severest water shortage areas in northwest China [1,2]. Xi'an, located in the Wei River Depression Basin (Figure 1), is the capital of Shaanxi Province, and was the location of 13 ancient dynasty capitals at various times in history. The city is bounded by the Chan River and Ba River to the east, the Feng River to the west, the Wei River to the north, and the Qinling mountains to the south (Figure 2). It has been subjected to serious land subsidence and ground fissure disasters since the 1960s [1–3]. Over the past decades, alongside the economic development and urban expansion that has occurred there, the range and magnitude of the detected subsidence zones have gone through four

stages. Historical leveling measurements show that, up to 1996, an area of over 150 km$^2$ had cumulative subsidence rates of more than 100 mm, with the maximum subsidence rate reaching 300 mm/a, and the maximum cumulative subsidence reaching over 2 m [4]. A total of seven main subsidence zones were formed in the southern, eastern, and southeastern suburbs of Xi'an city. After the 2000s, four land subsidence zones remained in the southern, southeastern, and southwestern suburbs of Xi'an city. The maximum land subsidence rate was found in the Xi'an Hi-Tech Zone in the west of Xi'an, where it reached 140 mm/a in 2012 [5]. Long-term intense groundwater extraction has caused severe land subsidence, and has triggered the formation of ground fissures in Xi'an. Fourteen approximately parallel ground fissures have emerged throughout the city in an east-north-east (ENE) direction. The ground fissures have controlled the land subsidence area, causing the subsidence areas to stretch into elliptical shapes with their long axes parallel to the fissure direction [4,5]. The consequences of land subsidence and ground fissures include degradation of the aquifer system and damage to utility infrastructure, such as buildings, railroads, highways, bridges, and subways [5–8].

Many researchers have conducted ground subsidence and fissure monitoring since 1959, and have obtained valuable measurements using conventional ground-based techniques, including leveling and GPS over Xi'an [9–12]. Although the traditional methods provide good precision of measurements and valuable deformation history, they are of limited use in the detection of land subsidence associated with groundwater depletion over large areas, and in retrieving data on the historic evolution of land subsidence and ground fissures, due to the labor-intensive and time-consuming nature of these problems. Since the 1990s, Interferometric SAR (InSAR) has become a revolutionary tool for measuring the surface deformation induced by various hazards [13–15]. To overcome the limitations of conventional InSAR, including decorrelation and atmospheric artifacts, advanced time series InSAR techniques, such as Persistent Scatterers InSAR (PSInSAR) [16], Small Baseline Subset (SBAS) InSAR [17,18], Temporally Coherent Point InSAR (TCPInSAR) [19], Quasi-PS (QPS) [20], Interferometric Point Target Analysis (IPTA) [21], and Intermittent SBAS (ISBAS) [22], have been proposed for different deformation scenarios. These advanced time series techniques have facilitated land subsidence monitoring in many urban areas, such as in Beijing [23], Shanghai [24], Wuhan [25], Taiyuan [26], Mexico [27], Italy [28,29] and others. In Xi'an, Zhang et al. (2009) studied the spatiotemporal evolution and mechanism of land subsidence and ground fissure activities from 1992–2006, using both GPS and InSAR observations [10]. Zhao et al. (2008, 2009) studied the evolution of land subsidence from 1992–2007 by using 15 ERS-1/2 and Envisat data, and analyzed land subsidence from 2005–2006 by utilizing 7 Envisat images [30,31]. Subsequently, Qu et al. (2014) studied the spatiotemporal evolution of land subsidence and ground fissures from 2005–2012 using multiple SAR datasets, as well as leveling, GPS, and groundwater data [5].

With the expansion of Xi'an city, land subsidence and ground fissures have developed continuously, which poses a serious threat to urban construction and infrastructure. Therefore, the study of the surface deformation in Xi'an is of great significance. We focus on the surface deformation in Xi'an city with the area of 39 × 44 km$^2$, outlined in Figure 1 by the red rectangle, where severe land subsidence and ground fissures have occurred in past several decades and few new results available after 2012. Time series InSAR techniques are applied to multi-band SAR data stacks to quantify the entire land subsidence phenomenon from 2012 to 2018. Subsequently, the spatiotemporal evolution of land subsidence in four regional areas is interpreted in detail. Moreover, the deformation characteristics along the Xi'an subway lines are explored for the first time. Finally, the possible underlying mechanics of land subsidence in Xi'an are discussed according to the InSAR-derived results.

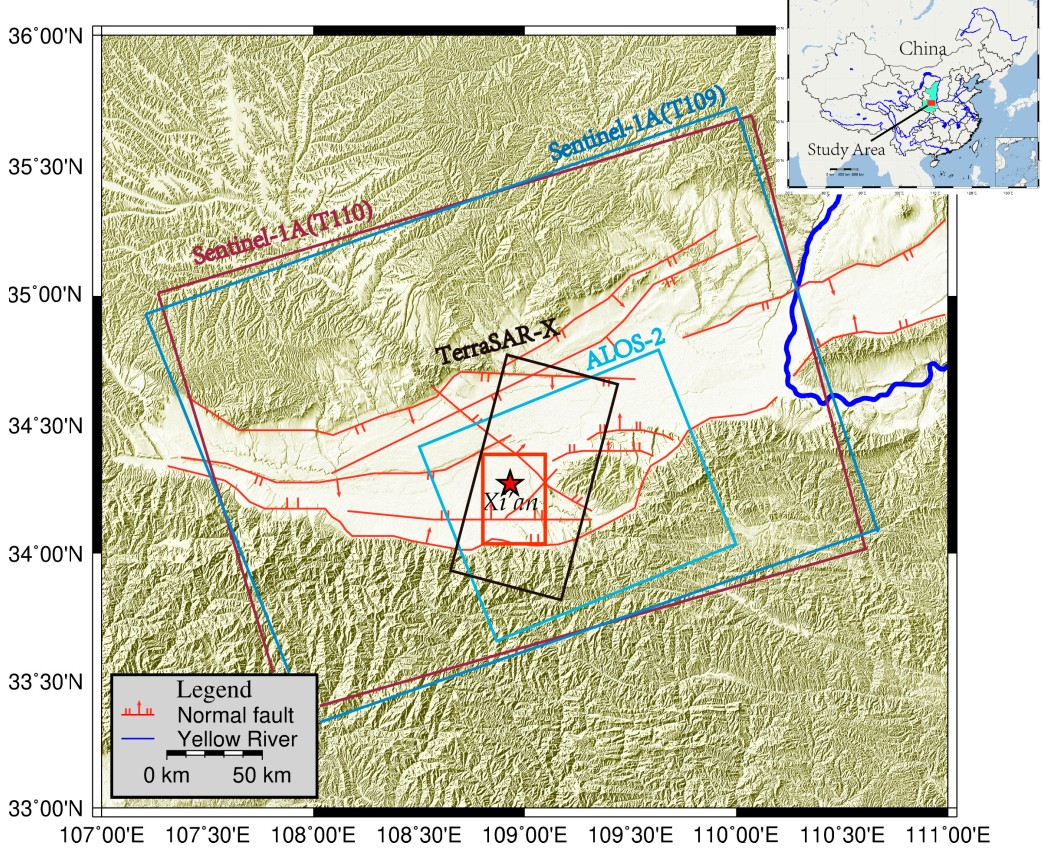

**Figure 1.** Study area and SAR data coverage used in this study. Different SAR tracks are represented by solid boxes with different colors. The red rectangle indicates the study region of Xi'an.

The paper is organized as follows. The geological background of the study area is discussed in Section 2. Section 3 presents the multi-datasets and data processing techniques involved. The results and accuracy validation are described in Section 4. Some analyses of regional land subsidence are given in Section 5. Discussions are provided in Section 6, and concluding remarks are given in Section 7.

## 2. The Geological Background of the Study Area

Xi'an city is located to the north of the Qinling mountain and south of the Loess Plateau. The elevation of the study area varies from 360 to 750 m. It belongs to a temperate, semi-humid continental monsoon climate, with an average annual temperature of 12 °C and an annual precipitation of about 585 mm [5].

Figure 2 is the shaded relief map of Xi'an, on which a Quaternary geology map is superimposed. The overall terrain of Xi'an gradually inclines from the northwest to the southeast, and the landform gradually transforms from flood plain to the loess tableland terraces. The range landform has caused the engineering geological conditions to exhibit banded inhomogeneity. Loess ridges and depressions are interchangeably distributed in central urban areas, where land subsidence and ground fissures have occurred.

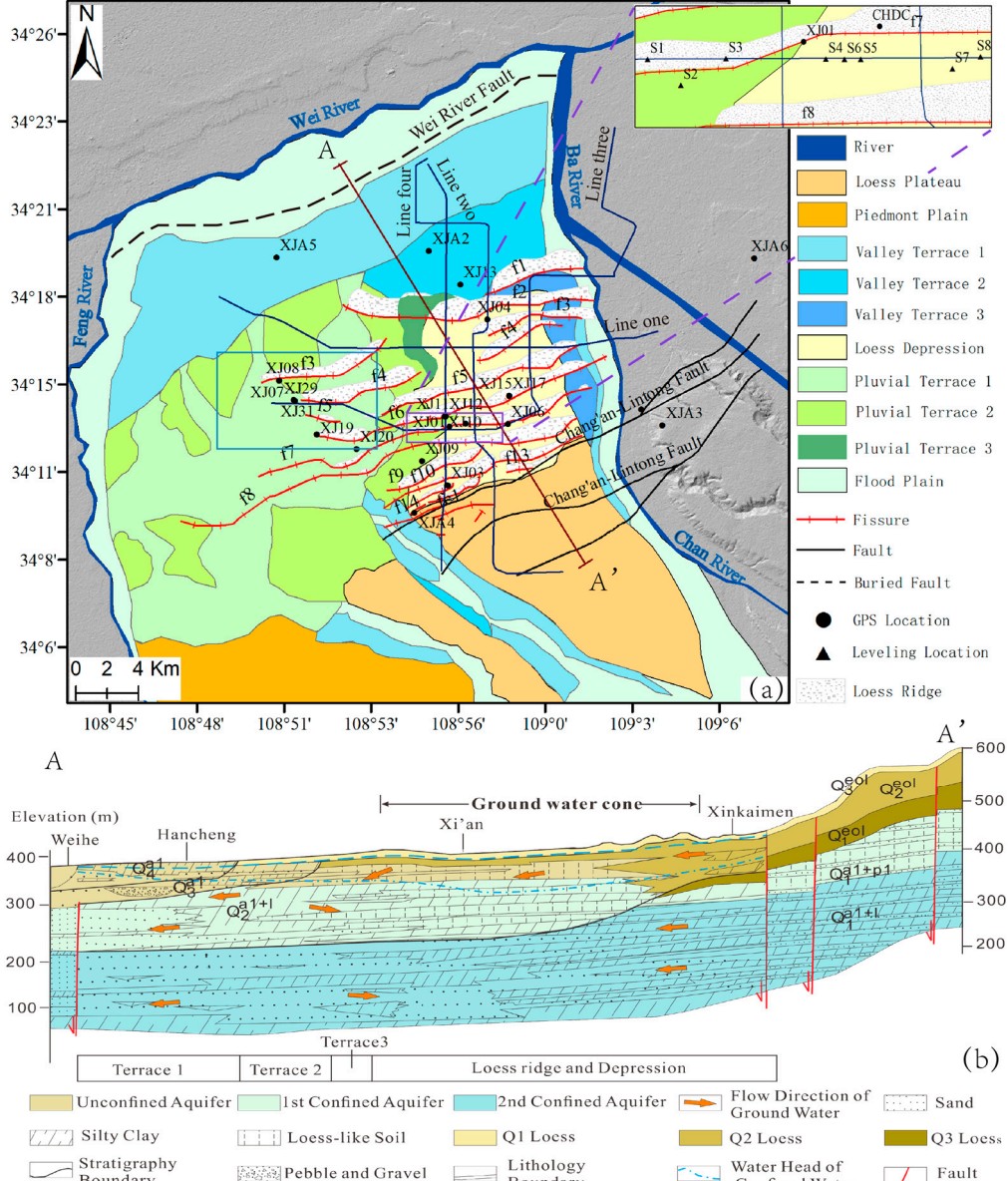

**Figure 2.** (**a**) Quaternary geology map, the inset is the distribution of leveling benchmarks along subway line three. The blue solid lines indicate the operated subway lines; Loess ridge areas are labeled with white blocks. (**b**) Hydrostratigraphic section along AA' in (**a**).

A series of normal faults have developed around the city, including the Chang'an–Lintong fault (CAF hereinafter), the Chan River fault, the Ba River fault, and the Wei River fault, as well as their secondary faults—among which the CAF fault (mainly in the ENE direction) is the most active one [32,33]. Research shows that neo-tectonic movement is not the primary cause of the local land subsidence [34].

Quaternary sediments in Xi'an are composed of a rich aquifer system, as shown in Figure 2b, which can be divided into three aquifers: an upper unconfined aquifer (less than about 100 m in depth); a middle confined aquifer (approximately 100–300 m in depth); and a lower confined aquifer (deeper than about 300 m, and also called the deep confined aquifer) [34]. The upper unconfined aquifer mainly consists of loess, loess-like soil, and sand. The first and second confined aquifers are separated by the first aquitard layer and distributed widely, and are mainly composed of fluvial and alluvial deposits. The deep confined aquifer is separated from the second confined aquifer by the deep aquitard layer.

The thickness of quaternary deposits increases from southeast to northwest, with a maximum thickness of more than 700 m in Xi'an. The stratums related to groundwater exploitation and land subsidence are mainly located above a depth of 300 m, and consist of three layers from top to bottom: Pleistocene loess (15–30 m), Middle-Pleistocene fluvial and lacustrine strata composed of thin silt clay and silt (30–130 m), and thick Lower Pleistocene fluvial strata composed of gravel and cohesive soil interbeds (below 130 m) [3].

This complex geological setting profoundly controls the characteristics of land subsidence and subsequent ground fissures in the area [34]. The aforementioned landform, lithology, and hydrogeology all influence the distribution of land subsidence [3].

## 3. Datasets and Method

### 3.1. SAR Datasets

To reveal the land subsidence characteristics over Xi'an from 2012 to 2018, a total of 127 SAR images from three SAR sensors are involved, including 37 descending-track TerraSAR-X images acquired from May 2012 to May 2015, 81 ascending-track Sentinel-1A images acquired from June 2015 to November 2018 (29 for track 105 and 52 for track 109), and 9 ascending-track ALOS-2 images acquired from September 2014 to October 2017. The coverage of the three types of data is shown in Figure 1, and the detailed parameters are summarized in Table 1. External DEM data provided by NASA from Shuttle Radar Topography Mission-1 (SRTM), with a spatial resolution of 30 m, is used to simulate and remove the topographic phases [35].

**Table 1.** Basic parameters of the three SAR datasets.

| Sensor | TerraSAR-X | Sentinel-1A | | ALOS-2 |
|---|---|---|---|---|
| Band (wavelength in cm) | X (3.1) | C (5.6) | | L (23.6) |
| Incident angle (°) | 28.6 | 39.2 | | 40.5 |
| Slant range spacing (m) | 0.9 | 2.3 | | 4.2 |
| Azimuth spacing (m) | 2 | 14.1 | | 3.2 |
| Pass direction | Descending | Ascending | | Ascending |
| Track number | 13 | 110 | 109 | 143 |
| Number of scenes | 37 | 29 | 52 | 9 |
| Date period | 2012/05/12–2015/05/28 | 2015/06/20–2017/03/05 | 2017/04/10–2018/11/07 | 2014/09/06–2017/10/28 |

Sentinel-1A Interferometric Wide Swath (IW) Single Look Complex (SLC) products, used in this study, are gathered using the novel Terrain Observation with Progression Scans (TOPS) in azimuth SAR imaging technique [36]. The level-1 IW image is provided as three separate sub-swaths (IW1, IW2, IW3), and each of them consists of a series of bursts. Only the four bursts that fully cover our study area are used. The accuracy of the co-registration for Sentinel-1 TOPS InSAR processing in the azimuth direction must reach one thousandth of one pixel, which is much more stringent than that required for the stripmap products. A wide-area IW product is created with two steps; that is, debursting and merging [37]. The former processing concatenates the individual bursts from one sub-swath into a single debursting sub-swath, and then several sub-swaths are merged into one single wide-swath product. Precise Orbit Determination (POD) data released by the European Space Agency (ESA) are used for orbital refinement and phase re-flattening.

### 3.2. GPS and Leveling Data

A GPS network composed of 31 benchmarks (Figure 2) constructed in 2005 has been surveyed annually: six stations (labeled as XJA1–XJA6) are set as base stations, and the other 25 stations (labeled as XJ01–XJ25) are set as monitoring points [6–8]. Unfortunately, for various reasons, only 23 GPS points were measured in 2014 and 16 points in 2015. Additionally, eight second-order leveling observations were available along subway line three in 2016 (Figure 2a). The precision of the GPS vertical components and leveling points are about 5 mm and 2mm, respectively [10].

*3.3. Method*

3.3.1. SBAS-InSAR

The small baseline subsets (SBAS) method is employed to derive the surface deformation, where small baseline interferograms are generated to mitigate the effects of temporal decorrelation and spatial decorrelation [17,18]. First, differential interferograms are produced based on temporal and spatial baseline thresholds. Then, multi-looking and Goldstein filtering are applied to improve the signal-noise-ratio of the interferograms. Phase unwrapping with the Minimum Cost Flow approach (MCF) algorithm is followed to the coherent pixels for each wrapped interferogram.

The unwrapped phase at pixel $(x, r)$ in the $k$th differential interferogram can be written in the following form:

$$\delta\varphi_{(x,r)} = \varphi_{(x,r)}(t_B) - \varphi_{(x,r)}(t_A) \approx \delta\varphi_{(def,x,r)} + \delta\varphi_{(\epsilon,x,r)} + \delta\varphi_{(\alpha,x,r)} + \delta\varphi_{(n,x,r)}, \qquad (1)$$

where $\varphi_{(x,r)}(t_B)$ and $\varphi_{(x,r)}(t_A)$ represent the phases at SAR acquisition dates $t_A, t_B (t_A < t_B)$ respectively; $\delta\varphi_{(def,x,r)}$ is the deformation phase component corresponding to the target movement along the satellite line-of-sight (LOS) direction; $\delta\varphi_{(\epsilon,x,r)}$ is the topographic error; $\delta\varphi_{(\alpha,x,r)}$ is the phase signal induced by the difference in atmospheric path delay between two observations, and $\delta\varphi_{(n,x,r)}$ is the noise phase.

Atmospheric artifacts corresponding to the height are removed by polynomial fitting from the interferograms. Finally, the Singular Value Decomposition (SVD) method was applied to estimate the time series deformation for each coherent pixel.

3.3.2. Persistent Scatterer Candidate Selection and Regression Analysis

When the SAR data is sufficient—that is, more than 25 acquisitions—the time series deformation and mean deformation rates are derived with the interferometric point target analysis (IPTA) technique [21,38,39].

The average spectral diversity and backscattering intensity are taken as two criteria to extract point target candidates from the co-registered SLCs [40]. High-quality point targets will be retained, and low-quality point targets will be eliminated through the regression analysis. Then, differential interferometric phases at the point targets are calculated using the co-registered SLCs and DEM in the SAR coordinate system, which includes five main parts: the residual orbital phase, residual topographic phase, deformation phase, atmospheric phase, and noise phase. Afterwards, two-dimensional (2-D) regression analysis is performed, and reliable results relative to the given reference point can be expected. Furthermore, the estimation of the deformation rate and height errors can be improved step by step through each iteration, where a low standard deviation indicates high precision of the interferogram. The nonlinear deformation is retrieved through low-pass filtering in the spatial domain, and followed by high-pass filtering in the temporal domain. Finally, time series deformation at each point target is obtained by adding the nonlinear deformation component to the linear deformation component [26].

The available external data (e.g., GPS and leveling observations) is limited, and there is no temporal acquisition overlap between ascending and descending data, making it difficult to derive multidimensional deformations. Additionally, vertical motion dominates the deformation field in Xi'an [5]. Therefore, all the SAR LOS measurements are projected into the vertical direction with respect to the corresponding incidence angles as follows.

$$d_{vertical} = \frac{d_{los}}{cos\theta} \qquad (2)$$

where $d_{vertical}$ is the vertical deformation; $d_{los}$ is the displacements in LOS direction; $\theta$ is the incidence angle.

## 4. Results and Validation

### 4.1. Deformation Rates

A total of 754 interferograms are constructed, including 287 from TerraSAR-X, 37 from ALOS-2, 131 from Sentinel-1A (T110), and 299 from Sentinel-1A (T109), to monitor the land subsidence in Xi'an from 2012 to 2018.

The vertical deformation rate maps over Xi'an region are generated independently from four different datasets and shown in Figure 3, where Figure 3a is the deformation map from 2012 to 2015 calculated with descending X-band TerraSAR data, Figure 3b is the deformation map from 2014 to 2017 calculated with ascending L-band ALOS-2 data, and Figure 3c,d are the deformation maps from 2015 to 2018 calculated with ascending C-band Sentinel-1A of Track110 and Track 109, respectively. All results are referred to the stable point labelled with the black star in Figure 3a.

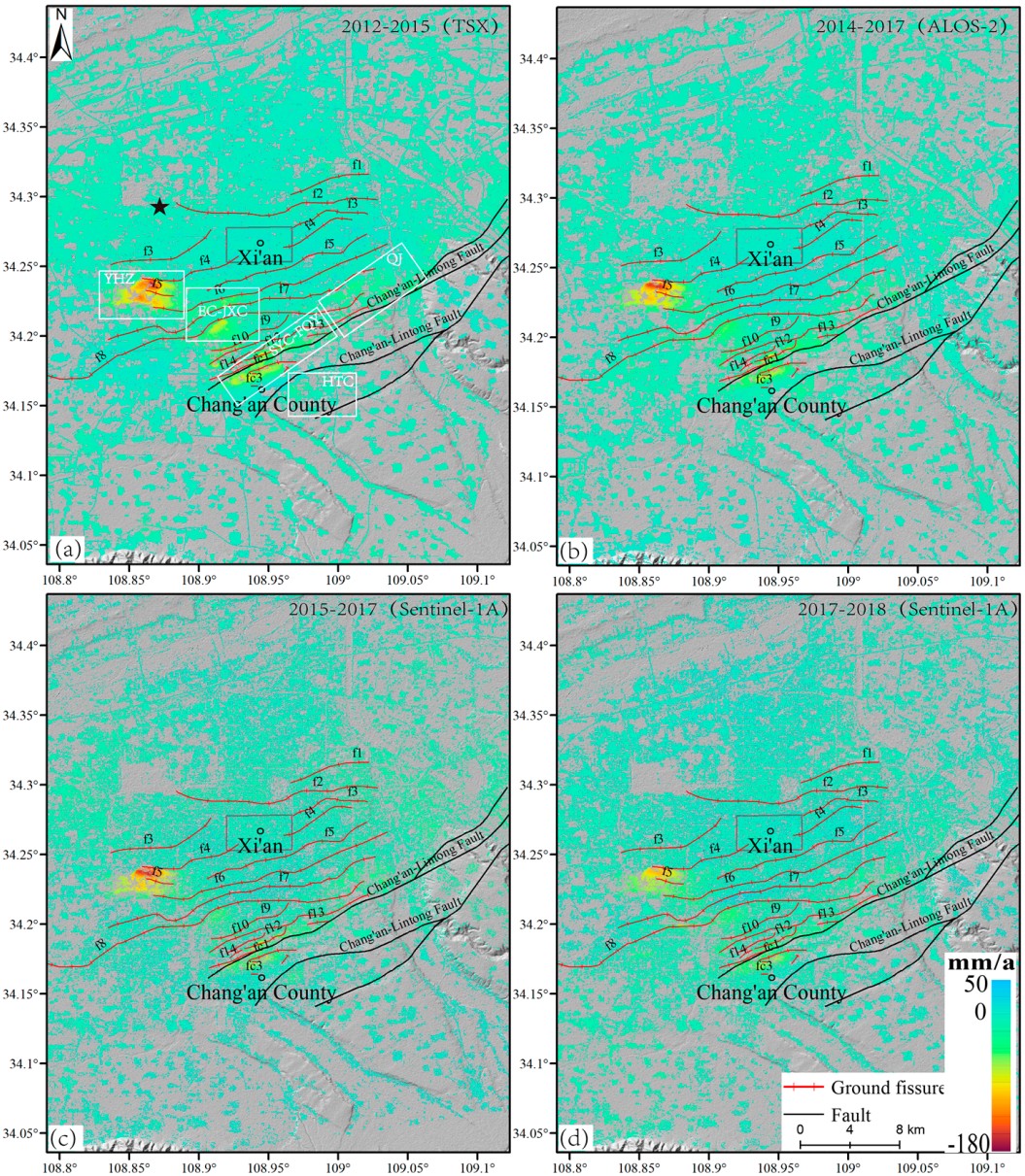

**Figure 3.** Average vertical deformation rate maps of Xi'an from 2012 to 2018, derived from TerraSAR-X (**a**), ascending ALOS-2 (**b**) and Sentinel-1A Satellite (**c,d**). Positive values in blue represent movement uplift, and negative values in red represent subsidence. The black star in (**a**) indicates the reference point.

During the whole monitoring period, the land subsidence zones were mainly concentrated in the southwest and southeast suburbs of Xi'an city. The whole spatial distribution of the subsidence centers has undergone no significant change in recent years. The largest land subsidence occurs at Yuhuazhai (YHZ hereinafter), with a maximum subsidence rate of around −175mm/a. Sanyaocun-Fengqiyuan (SYC-FQY hereinafter) is the second largest subsidence zone, with a deformation rate ranging from −20 to −90 mm/a. These severe subsidence zones have caused some impacts on the operation of subway lines, which will be further analyzed in Section 5.2. It is worth noting that slight uplifting zones are detected from four different SAR datasets from 2014 to 2018, which will be further discussed in Sections 5.1.2 and 6.3.

## 4.2. Validation of InSAR Results

Independent InSAR observations that cover the same area and a similar time interval are utilized to cross-validate the InSAR-derived surface deformation measurements. To do so, all deformation rate maps are first resampled to the same image resolution, and the YHZ subsidence bowl highlighted by a blue rectangle in Figure 2a is taken as the validation region.

Scatter plots of three SAR datasets are presented in Figure 4, where Figure 4a shows the linear fitting model between the TerraSAR-X and ALOS-2 datasets with a root-mean-square error (RMSE) of about 13.2 mm, and Figure 4b shows the linear fitting model between the Sentinel-1A and ALOS-2 datasets with a RMSE of about 6.9 mm. High consistency can be achieved with different SAR data, while the discrepancy can be explained by the inconsistent monitoring period, different SAR scenes, and slight horizontal displacement [5].

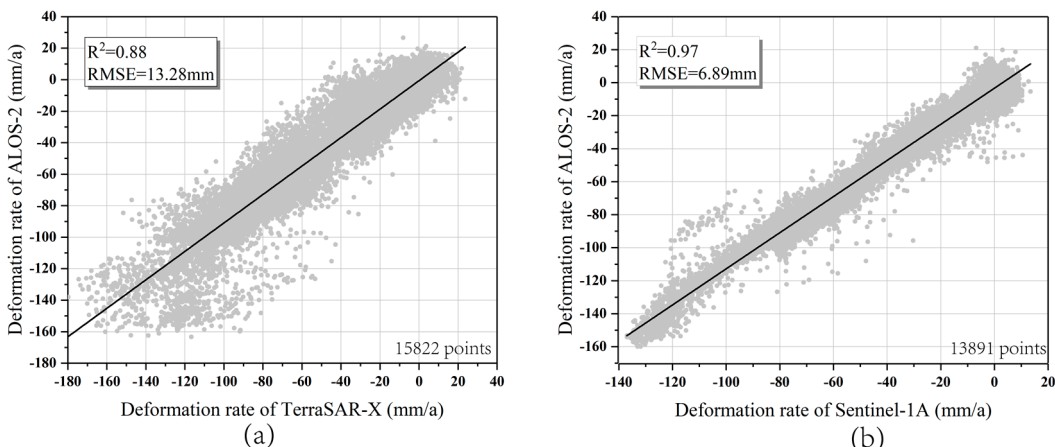

**Figure 4.** Scatter plots of the vertical deformation rates among three SAR datasets. (**a**) Between ALOS-2 and TerraSAR-X datasets, (**b**) between Sentinel-1A and ALOS-2 datasets.

## 4.3. Calibration of InSAR Results

A calibration of the InSAR-derived deformation results is performed with GPS and leveling measurements. The locations of GPS and leveling benchmarks are shown in Figure 2a. The comparisons with GPS measurements in 2014 and 2015 are shown in Figure 5a,b, respectively, while measurements from 2016 are compared with eight leveling measurements, as shown in Figure 5c. InSAR measurements from TerraSAR-X datasets at each GPS benchmark are calculated within a square of 50 m × 50 m. The standard deviation within the square is calculated and shown as the error bar in Figure 5a,b. The standard deviations of the differences are 9.83 mm and 6.09 mm in 2014 and 2015, respectively. Additionally, a significant difference between InSAR and GPS measurements at some GPS benchmarks, such as at XJ12, XJ19 and XJA4 in 2014, and CHDC, XJ25 and XJA6 in 2015 can be seen in Figure 5a,b. These can be explained by the localized deformation, the horizontal deformation effect, and the low GPS vertical accuracy under a poor measurement environment.

Leveling benchmarks were mounted for the monitoring of subway line deformation during the construction of subway line three from 2012 to 2016. Both leveling and InSAR measurements from ALOS-2 datasets uncovered surface rebound in 2016, with a standard deviation of 2.2 mm (Figure 5c), which will be discussed later.

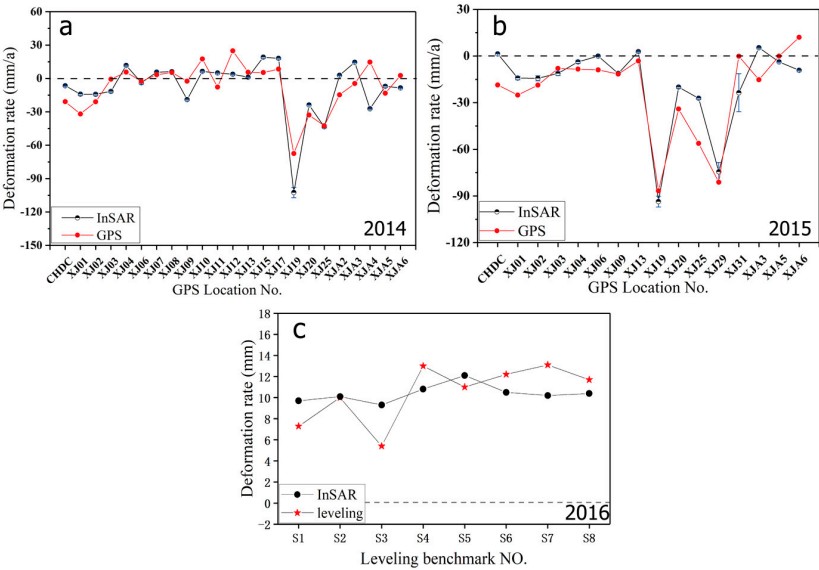

**Figure 5.** Comparison of InSAR measurements with GPS and leveling measurements. (**a**) With GPS measurement in 2014; (**b**) with GPS measurement in 2015; (**c**) with leveling measurement in 2016. The error bar, with one standard deviation, is shown in (**a**,**b**) at each GPS benchmark.

## 5. Analysis

### 5.1. Regional Land Subsidence Characteration

Five subsidence zones were detected from 2012 to 2018 based on the multi-band SAR data shown in Figure 3, including YHZ, Qujiang residential district (QJ hereinafter), SYC-FQY, Electronic Mall-Jixiangcun (EM-JXC hereinafter), and Hangtiancheng (HTC). Compared with previous research [5], the HTC subsidence zone was formed in 2012 and the subsidence rate varied from about −30 mm/a to 0 mm/a, while the other four land subsidence centers were formed as early as 2005.

Three major land subsidence areas were apparent at YHZ, EC and SYC-FQY, which will be analyzed in detail later, while the average deformation rate of the QJ subsidence area was also approximately –30mm/a. Cumulative time series vertical deformation of four feature points (P1, P2, P3 and P4) at three subsidence centers from TerraSAR-X, ALOS-2, Sentinel-1A (T110/109) data stacks are extracted. The earliest data acquisition date is set as the reference date, and the linear interpolation of the former dataset is set as the reference deformation of the later temporally overlapped dataset. That is, the reference date between TSX and ALOS-2 is set at September 6, 2014, between Sentinel-1A (T110) and ALOS-2 is set at 27 June 2015, and between Sentinel-1A (T110) and Sentinel-1A (T109) is set at 5 March 2018. Furthermore, the spatiotemporal evolution of these three land subsidence centers and ground fissures are analyzed with four profiles.

#### 5.1.1. Land Subsidence in YHZ

The YHZ subsidence bowl, the largest subsidence center in Xi'an city since 2005, is located in the southwestern suburb of Xi'an and has caused severe damage to buildings, and railroads. It has the potential to impact the safety of subway line three.

Figure 6a–d shows the different land subsidence rates from different SAR datasets. It can be seen that the land subsidence is located between ground fissures f4 and f6 in the spatial distribution. The detected deformation rate ranges from −180 mm/a to 10 mm/a. It can also be seen from Figure 6a

that there were two subsidence bowls, which were separated by ground fissure f5. The one between ground fissures f5 and f6 decreased significantly from −143 mm/a to −60 mm/a. As for the one between ground fissures f4 and f5, the maximum deformation rate varied between −140 mm/a and −160 mm/a before 2017, and decreased to −110 mm/a in 2018.

In particular, Figure 6e shows the Google Earth image over the subsidence district, which indicates the location and high density population in the YHZ bowl area; blue color indicates the industrial and residential zones. The over-withdrawal of groundwater is the main cause of land subsidence, and this will be quantitatively analyzed in Section 6.3.

The cumulative time series of land subsidence at point P1 during the entire observation period is shown in Figure 6f, which is labeled with a red dot in Figure 6a. The cumulative subsidence reached 820 mm from 2012 to 2018.

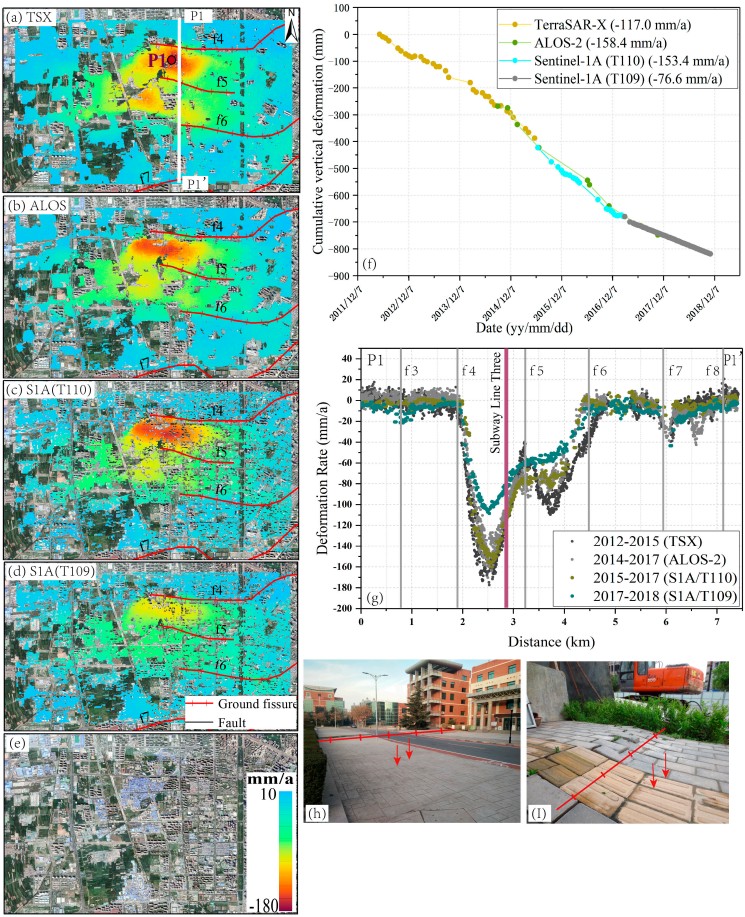

**Figure 6.** Land subsidence in the YHZ subsidence bowl. (**a**) From 2012 to 2015 with the TerraSAR-X dataset; (**b**) from 2014 to 2017 with the ALOS-2 dataset; (**c**) from 2015 to 2017 with the Sentinel-1A (T110) dataset; (**d**) from 2017 to 2018 with the Sentinel-1A (T109) dataset; (**e**) Google Earth image; (**f**) cumulative time series deformation of P1; (**g**) annual deformation rate from different-band datasets along the profile of P1P1′ from 2012 to 2018, whose position is marked in (**a**); (**h,i**) the field investigation photos of ground fissures.

### 5.1.2. Land Subsidence in the EC-JXC District

Figure 7a–d shows the different land subsidence rates from different SAR datasets over the EC-JXC subsidence zone. Two profiles are extracted along the lines P2P2′ and P3P3′ (see Figure 7a) to analyze the evolution of surface deformation, and this is shown in Figure 7g,h, respectively. The subsidence rate of the EC subsidence zone significantly slowed down year by year, and changed in both the space and time domains during the observation period (Figure 7a–d,g). At the feature point P2, the land

subsidence rate varied from about −70 mm/a to −40 mm/a, and the accumulated land subsidence reached −345 mm, as shown in Figure 7f.

Figure 7a shows that the JXC subsidence zone is located between fissure f6 and f7, which is another urban village with a highly dense population and groundwater consumption. The subsidence rate varied from −20 mm/a to −50 mm/a, observed from the TerraSAR-X datasets, and then rebounded to about 5 mm/a after 2015 due to a restriction on ground water pumping.

Additionally, surface uplift is detected in the area between ground fissure f7 and f8 (Figure 7a–d). According to multi-band InSAR-derived monitoring results, the subsidence rate was relatively high (−50 mm/a) between 2012 and 2015, and then began to rebound year by year, and the surface deformation rate increased to 20 mm/a between 2017 and 2018. The deformation rate remains constant in the southern section of ground fissure f8.

The same rebound phenomenon is uncovered in Changzhou, Jiangsu Province, where the uplift of the land surface is mainly due to strict regulatory measures to prohibit the exploitation of groundwater [41].

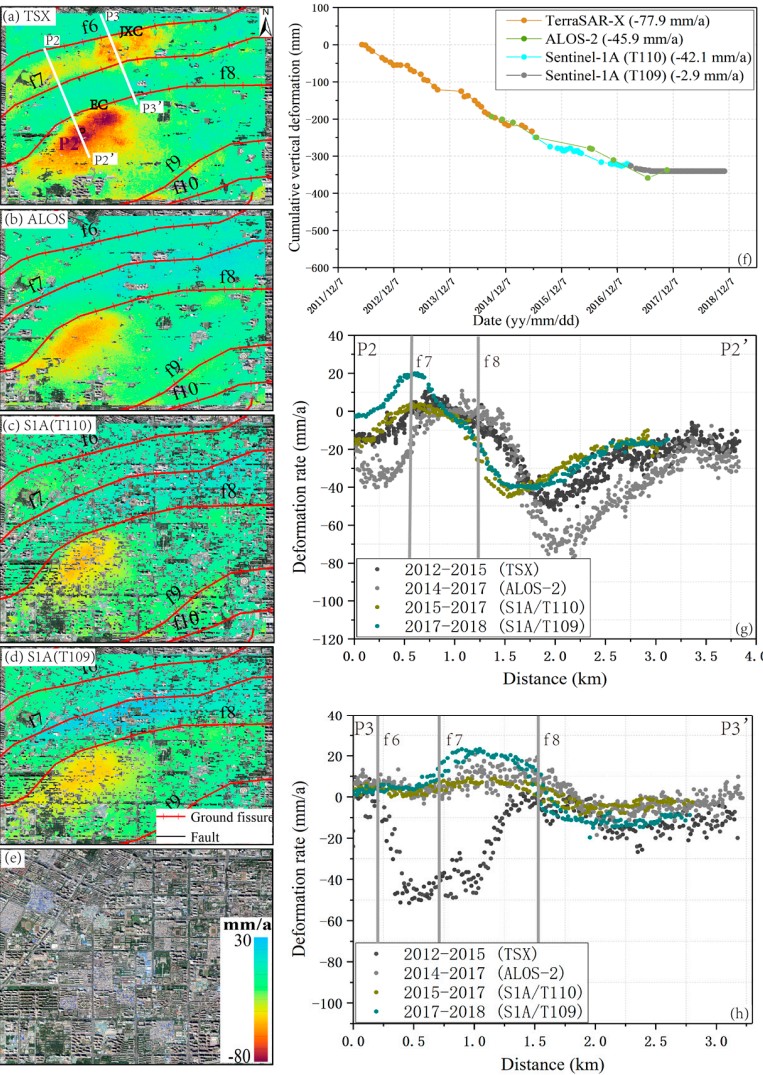

**Figure 7.** Land subsidence in the EC-JXC area, located in the south-west sector of Xi'an city. (**a**) From 2012 to 2015 with the TerraSAR-X dataset; (**b**) from 2014 to 2017 with the ALOS-2 dataset; (**c**) from 2015 to 2017 with the Sentinel-1A (T110) dataset; (**d**) from 2017 to 2018 with the Sentinel-1A (T109) dataset; (**e**) Google Earth image; (**f**) cumulative time series deformation of P2; (**g,h**) annual deformation rates from different-band datasets along the profiles P2P2′ and P3P3′ from 2012 to 2018, respectively, whose positions are marked in (**a**).

### 5.1.3. Land Subsidence in the SYC-FQY District

Figure 8a–d shows the different land subsidence rates from different SAR datasets over the SYC-FQY subsidence zone, which is situated in the south-west sector of Xi'an city and consists of two main subsidence areas. The SYC subsidence center is between ground fissure f11 and the CAF fault, and the FQY subsidence center is between the CAF fault and fissure fc4. The deformation rates of the two subsidence zones range from −120 mm/a to −30 mm/a from 2012 to 2018, with an average deformation rate of about −70 mm/a.

Figure 8h shows the deformation changes along line P4P4′ from 2012 to 2018, where two subsidence zones can be clearly seen. The FQY subsidence center decreased obviously from 2012 to 2018, while the SYC subsidence center decreased significantly in 2017 and 2018. Cumulative deformation time series at feature points P3 and P4 are extracted in two subsidence zones, as shown in Figure 8f,g. The accumulated land subsidence reached −450 mm at P3, and −510 mm at P4.

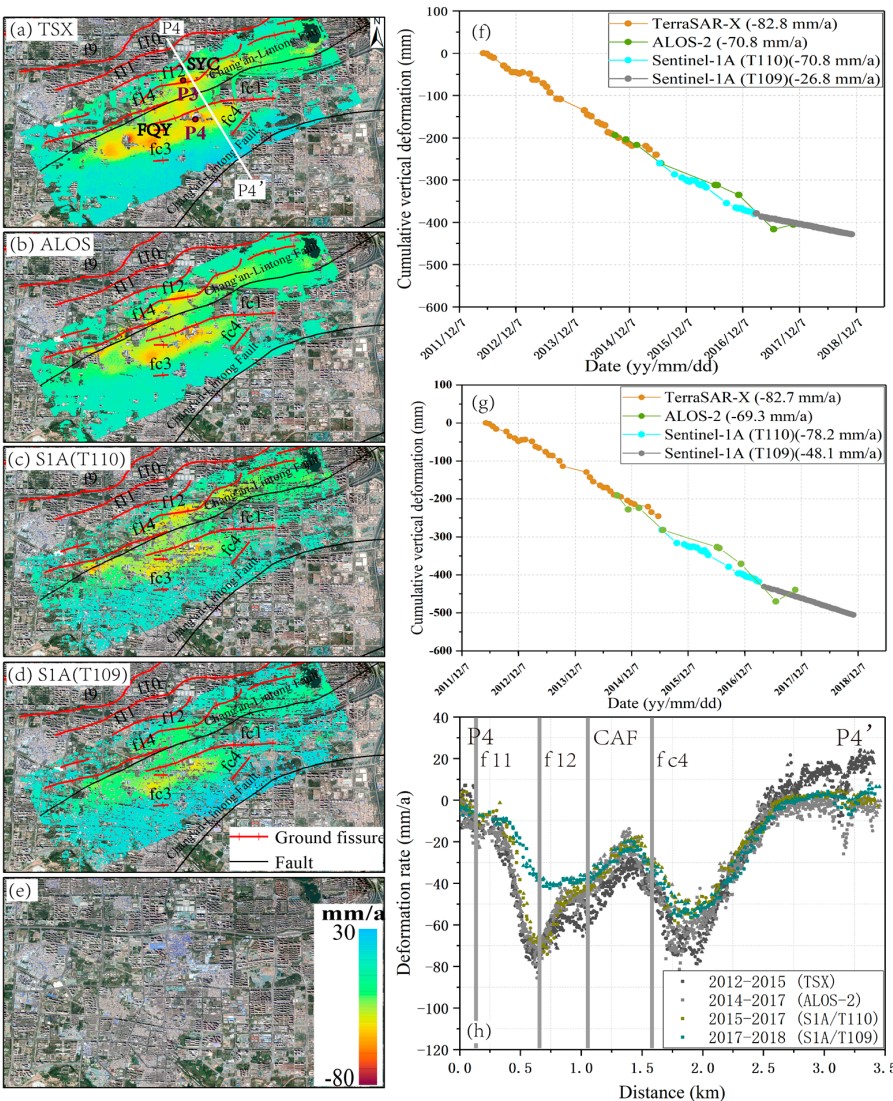

**Figure 8.** Land subsidence in the SYC-FQY area, located in the south-west sector of Xi'an city. (**a**) From 2012 to 2015 with the TerraSAR-X dataset; (**b**) from 2014 to 2017 with the ALOS-2 dataset; (**c**) from 2015 to 2017 with the Sentinel-1A (T110) dataset; (**d**) from 2017 to 2018 with the Sentinel-1A (T109) dataset; (**e**) Google Earth image; (**f**,**g**) cumulative time series deformations at the feature point P3 and P4, respectively, whose positions are marked in (**a**); (**h**) annual deformation rates from different band datasets along the profile P4P4′ from 2012 to 2018, whose position is marked in (**a**).

### 5.2. Land Subsidence along Subway Lines

The construction and operation of the subway is greatly affected by land subsidence and ground fissure deformation. By the end of 2018 in Xi'an, a total of four subway lines (lines one, two, three and four) were in operation, and five subway lines are currently under construction. The deformation of the whole subway network within a width of 1 km is monitored by Sentinel-1A data, and is shown in Figure 9. Subway line one passes through downtown from east to west, where the land deformation was relatively stable. Line two, three and four have suffered from different land subsidence rates, and ground fissures. To analyze the deformation along the subway lines more concretely, the deformation rates along three lines within a width of 1 km are extracted separately, and are shown in Figure 10.

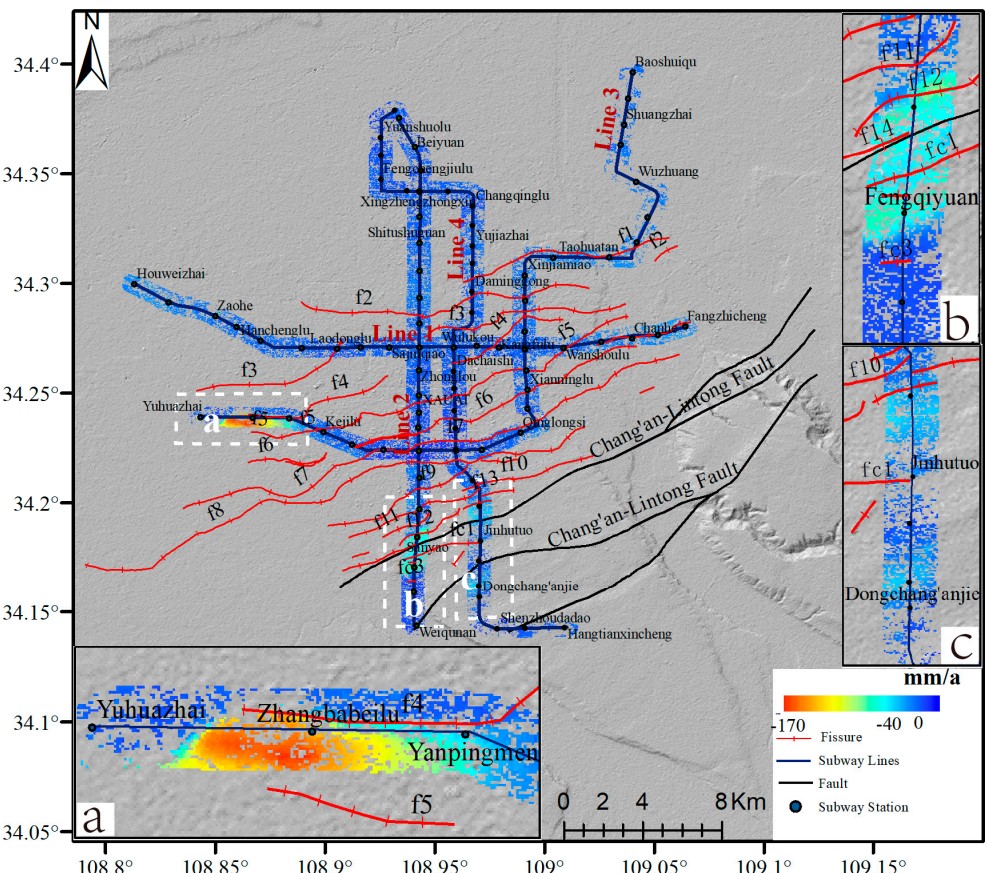

**Figure 9.** Deformation rate map along four subway lines (line one, line two, line three and line four) derived by the Sentinel-1A datasets in 2018. Regions a, b and c, marked with white dashed rectangles, are three main subsidence areas along line three, line two and line four, and their enlarged figures are all superimposed.

Subway line two passes through downtown from north to south, the first phase of which started operating at the end of 2011. The deformation along it is quite stable. The second phase of line two is in the southern part of the area, between Huizhanzhongxin and Weiqunan station, and started operating in April 2014. Two subsidence areas were distributed around the south end of subway line two, as indicated by the white dashed boxes and in inset (b). The maximum subsidence rate reached 60mm/a shown in Figure 10a, which was located in the section between the Huizhanzhongxin station and FQY station.

Subway Line three started excavating completely in 2012 and operating in November 2016. Land subsidence was concentrated in the section between the Yanpingmen and Yuhuazhai stations as indicated by the white dashed boxes and in inset (a) in Figure 9. The maximum subsidence rate reached –145 mm/year (Figure 10b) near Zhangbabeilu station. The subway line three passes through

the YHZ subsidence bowl, and the annual maximum deformation rate reaches -150 mm/a viewed from a cross-section (Figure 6g). The north and south sides of the subway rail have uneven subsidence rates and large deformation gradients, which affects the smoothness of the route and the safety of subway operations.

Subway line four, the latest subway line in Xi'an, opened to the public at the end of 2018. Subsidence occurred in the segment from Datangfurongyuan to Shenzhoudadao station, with the subsidence rate ranging from 0 mm/a to −35 mm/a (Figure 10c), passing through fissures f8, f9, and f10, and the CAF fault.

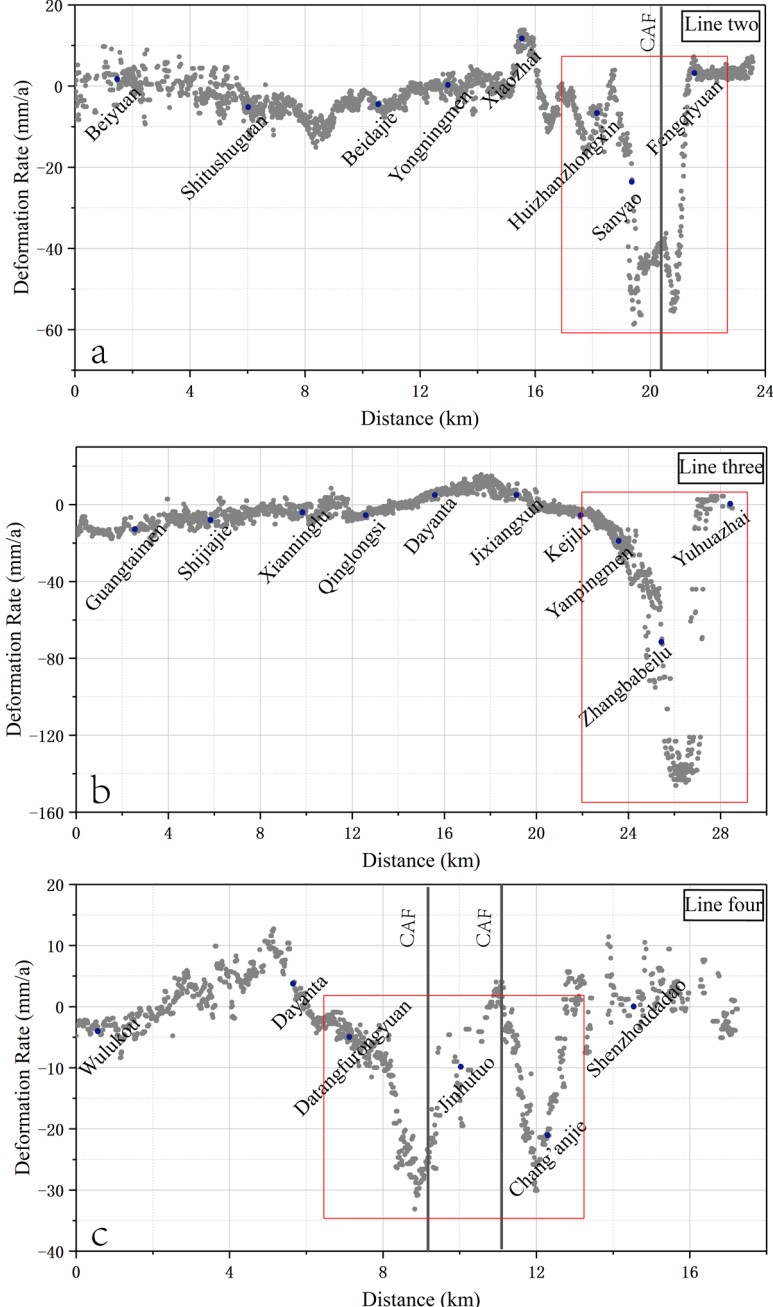

**Figure 10.** Deformation rate profile along subway line two (**a**), line three (**b**) and line four (**c**); the solid blue points denote the subway stations. The superimposed red rectangle boxes represent the maximum subsidence zone along the subway lines. The CAF is indicated by the grey line.

## 6. Discussion

The distribution of land subsidence in Xi'an is mainly attributed to excessive exploitation of groundwater, and is also influenced by several other factors, including geology characteristics, and ground fissures.

### 6.1. Correlation between Land Subsidence and Geology Characteristics

The spatial pattern of land subsidence is fundamentally controlled by stratigraphic formation and geological structures. As the loess ridges and depressions are intersectionally distributed in Xi'an (see Figure 2), and because of the differential thicknesses of compressible sediments deposited in adjacent blocks, all the subsidence zones are located in the loess depression section, as shown in Figure 3.

In addition, Xi'an is a pluvial terrace plain, and all Pleistocene loess, Middle-Pleistocene and Lower Pleistocene are the main underground water exploitation strata. As can be seen in Figures 2a and 3, the land subsidence regions, including YHZ, EC-JXC, and QJ, are mainly located in the pluvial terrace, and controlled by normal faults. Only the SYC-FQY subsidence region is located in the loess plateau, but it corresponds to different Quaternary geology, and the sedimentation characteristics are different, which results in a different compaction when the water-level declines.

Figure 2b depicts the hydrostratigraphic section along profile AA'. Land subsidence in Xi'an is largely caused by compaction of cohesive soils (including interbeds). When the groundwater level rises, elastic deformation occurs in sandy soil area and results in the rebound of the land surface. However, the consolidation of the aquifer system of clay soil is mostly irreversible. As the hydrostratigraphic characteristics at the YHZ, SYC-FQY, and QJ subsidence areas are different, the land subsidence magnitudes are different when the local groundwater heads decrease by the same magnitude [42]. Hence, the different magnitudes of land subsidence in the spatial domain are likely caused by the different thicknesses and hydraulic properties of the aquifers and aquitards underlying these areas.

### 6.2. Correlation between Land Subsidence and Ground Fissures

Previous researches show that CAF, one of the main active normal faults, controls the 14 ground fissures in the city of Xi'an. That is, 14 ground fissures are formed at the hanging wall of CAF [33,43,44].

Firstly, from the perspective of spatial distribution, the ground fissures and land subsidence are closely related. Ground fissures occur at the junctions between loess ridge and depression areas, and spread along the south edge of a loess ridge (Figure 2a) [34,44]. And land subsidence zones are formed and controlled by two adjacent ground fissures shown as elongated ovals or half-ovals, which can be seen from Figure 3. The annual deformation rate along the profiles P1P1', P2P2', P3P3' and P4P4', which is perpendicular to the regional ground fissures, are shown in Figures 6g, 7g,h and 8g, where there is great gradient of surface deformation crossing two sides of the fissures. It indicates that the activity of ground fissures is different at different locations.

Secondly, from the viewpoint of the activity and evolution, the ground fissures in Xi'an city are profoundly influenced by the groundwater extraction and land subsidence. Before the over-exploitation of groundwater, it was under natural conditions, i.e., the natural recharge was dynamically balanced by natural discharge, and the water levels were relatively stable. However, with the increasing demand of groundwater, more groundwater has been discharged from than recharged to the aquifer system, resulting in a substantial drop in confined aquifer water levels and forming a cone of groundwater depression. The groundwater flows toward the center of the pumping. Consequently, the subsidence zone is formed between the loess ridge and depression due to the different thicknesses of compressible sediments deposited in adjacent blocks. With the increase of land subsidence, the stress is concentrated on the preexisting fault plane buried by the Quaternary deposits. Once the stress concentration reaches the critical point, preexisting faults will be reactivated and begin

to slip, forming ground fissures at the ground surface. The detailed description can also be referred to some published materials [33,34,44].

Additionally, according to the gradient of land subsidence, the active ground fissures can be detected, as shown in Figure 11. The purple line in Figure 11b indicates the updated location of ground fissures f4 and f6, and the continuous extension to the west, which was verified by the field investigation and is consistent with the descriptions in a previous paper [43].

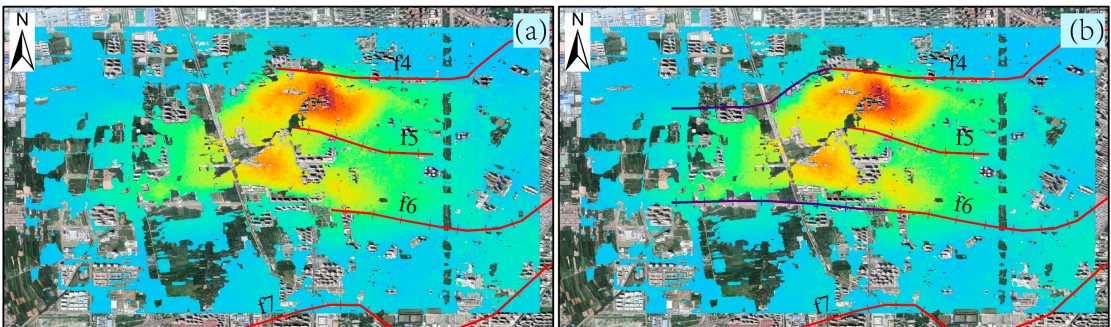

**Figure 11.** Updating the location of ground fissures. (**a**) The archived record of the location of fissures f4, f5 and f6; (**b**) the updated location of ground fissures f4 and f6, according to land subsidence.

### 6.3. Correlation between Land Subsidence and Underground Water Changes

Land subsidence in Xi'an was first discovered in the 1960s by a citywide level survey. The primary source of the land surface subsidence in the Xi'an area is attributed to the long-term and excessive withdrawal of groundwater [3].

Large amounts of data from the literature were analyzed to compare the correlation between land subsidence and the development of groundwater extraction in history [3,5,42,44]. The history of groundwater extraction in Xi'an can be generalized into three stages: the preliminary stage (before 1970), with a maximum deformation rate of less than 100 mm/a; the rapid development stage (1971–1997), with a significant increase reaching more than 190 mm/a; and the slow development stage (after 1997) following government intervention and the decrease in water wells after 1998. During its development, land subsidence generally followed the stages of groundwater extraction in the region. Based on ground-based techniques measurement data, land subsidence also went through three stages based on the subsidence rate: a slow subsidence stage (before 1970); a strong (accelerated, hyper and sustained) subsidence stage (1971–2001); and a decreasing subsidence stage (after 2001) [34,42,44].

Additionally, six groundwater recharge demonstration sites have been implemented since 2009 to accelerate the rise of the groundwater level. By 2011, the six recharging points had accumulated nearly 380,000 cubic meters of recharged water. According to the data analysis of the monitoring wells, the maximum rise of the underground water level in the recharging area is about 2 meters, especially near the recharging well of Dayan pagoda, with an average uplift of 2.11 meters [45]. Some areas that suffered from severe land subsidence before 2000, such as the Hujiamiao, Xinjiapo, Balicun and Xiaozhai-Dayan pagoda zones, have started to rebound. Figures 6e, 7e and 8e show that the five detected land subsidence zones comprise multiple urban villages with dense populations and relatively less development, and most domestic water is self-provided using wells by pumping groundwater. Further, in these backward urban villages, groundwater is still largely exploited [45], further emphasizing the close relationship between land subsidence and groundwater extraction in both time and space.

To better reveal the mechanism of the local land subsidence and explain the subsurface process, a flat lying sill model with distributed contractions embedded in a homogenous elastic half space is employed [46]. It has been widely and successfully used in modeling groundwater change-induced land subsidence, such as in Taiyuan [26], Yuncheng [47], and Texas [48]. To this end, the YHZ subsidence region, shown as the blue rectangle in Figure 2a, is taken as the inversion study area.

The deformation model is defined by the following six parameters: center coordinate X, Y, length, width, depth, and dip angle, shown in Table 2, which are determined by the constrained linear least-squares inversion procedure [49]. The patches of the distributed contraction sources are set as 0.15 km × 0.15 km. To identify the inversion depth of the flat-lying sill, the depth values—ranging from 0.01 m to 1000 m—are examined by calculating the root mean square (RMS) misfit of residuals (Figure 12). The optimal depth value is finally set as 120 m, with which the RMS misfit is the lowest one. Actually, the confined groundwater exploitation depth in the YHZ bowl is around 122.6 m, as reported by Xi'an Land and Resources [43].

**Table 2.** Geometric parameters used in the sill inversion model.

|  | X (km) | Y (km) | Length (km) | Width (km) | Depth (km) | Dip Angle (°) | Strike Angle (°) |
|---|---|---|---|---|---|---|---|
| Sill | 4 | 2.2 | 9 | 7 | 0.12 | 0 | 90 |

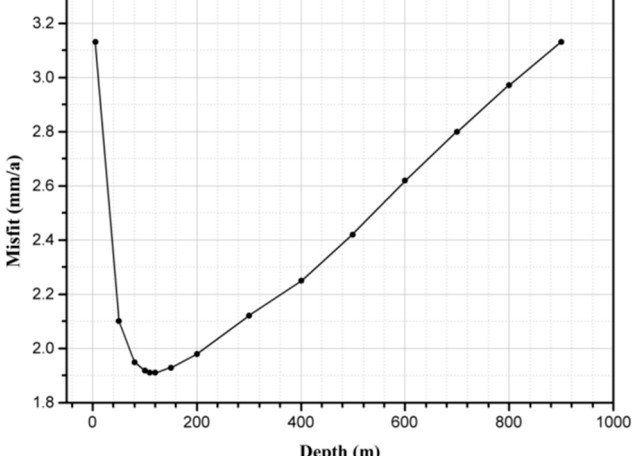

**Figure 12.** The variations of root mean square (RMS) of misfit with the depth of the flat-lying sill (black dots).

Figure 13 shows the observed, modeled, and residual deformation maps and profile of the observed and modeled deformation in 2017 along line P1P1'. It is clearly seen that the magnitude and shape of the modeled deformation matches well with the observed measurements. The root-mean-square error (RMSE) of the residual is chosen as the prediction-fit criterion, which is 1.98 mm for the optimal parameters. Good consistency can be seen in Figure 13d, where the correlation coefficient between the observed and modeled deformation reaches up to 99%. Finally, the inverted sill contraction distribution map is shown in Figure 14, where the maximum contraction at a depth around 120 m amounts to approximately 180 mm in the vertical direction.

The deformation model suggests that the contracted sill model is suitable for the land subsidence inversion; that is, the YHZ land subsidence center in 2017 can be explained by the excessive groundwater exploitation at a depth of around 120 meters.

Therefore, to improve the sustainable development of Xi'an city, some water resource utilization rules must be applied in the future, such as the restriction or prohibition of groundwater exploitation and a sustainable increase of the usage of surface water.

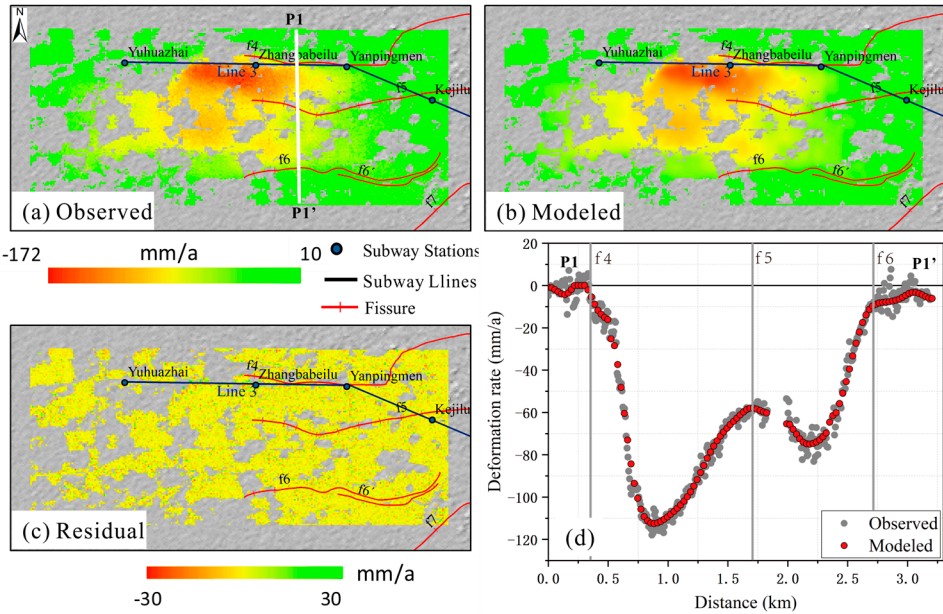

**Figure 13.** (**a**) The vertical deformation rate in 2017 over the YHZ subsidence bowl; (**b**) best-fitting model from a flat-lying distributed rectangle source in an elastic half-space; (**c**) residuals between the observed and modeled deformation; (**d**) deformation rate comparison between modeled and observed results along profile P1P1'.

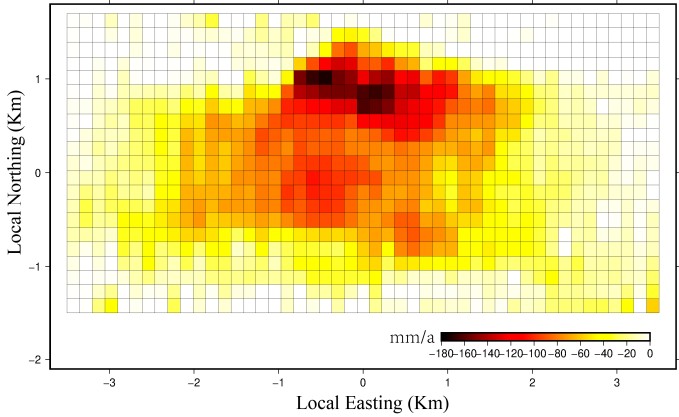

**Figure 14.** The vertical contractions of the inferred flat-lying sill model at a depth of 120 meters.

## 7. Conclusions

Multi-sensors and multi-band (X-, L- and C-band) SAR datasets were successfully applied to map the large-scale land subsidence over Xi'an city from 2012 to 2018. Four subsidence rate maps were generated from different SAR images acquired by the TerraSAR-X, ALOS-2 and Sentinel-1A (T110/T109) satellites. The main conclusions are summarized as follows:

InSAR deformation results were validated with independent SAR datasets and calibrated with GPS and leveling measurements. An accuracy of around 10 mm/a was achieved. Five main localized subsidence centers were detected from four InSAR measurements, and the spatiotemporal characteristics of three severe land subsidence zones were analyzed in detail. These land subsidence areas are mainly distributed in the southern, southeastern, and southwestern urban areas of Xi'an, among which the YHZ subsidence center is the largest subsidence zone in Xi'an city, with an accumulated subsidence over 6 years of about −820 mm. In addition, the slight uplifting in the EC-JXC was first detected by InSAR measurements from 2017 to 2018, with a maximum uplift rate of about 20 mm/a.

The surface deformation of the Xi'an subway network was accurately mapped with Sentinel-1A SAR datasets. Results indicate that the deformation along line one is relatively stable, while the deformation at the west section of line three, the south section of line two, and line four are suffering severe land subsidence and uneven deformation across local ground fissures. The maximum deformation rate along line three had reached −145 mm/a.

The correlations between land subsidence, geology formation, ground fissures, and groundwater exploitation were qualitatively and quantitatively analyzed. The study shows that excessive exploitation of underground water was the main cause of land subsidence, while CAF faults and ground fissures controlled the location and extension of Xi'an land subsidence.

Systematic research on city land subsidence was undertaken with multi-sensor SAR datasets, which will be able to directly assist decision-making on land subsidence mitigation and water resource management. Unfortunately, the lack of groundwater head data makes it difficult to quantitatively assess the relationship between groundwater exploitation and land subsidence changes. In future, it will be necessary to collect various in-situ measurements, such as the groundwater change level and extensometer data to better understand the complex subsurface processes and further confirm the mechanism of land subsidence, and the temporal evolution of land subsidence.

**Author Contributions:** All authors participated in editing and reviewing the manuscript. M.P. implemented the methodology, analyzed InSAR results and made these figures and wrote the original paper. C.Z., Q.Z. and Z.L. (Zhong Lu), Z.L. (Zhongsheng Li) designed the research program, supervised the research, and revised manuscript.

**Funding:** This work is jointly supported by the National Natural Science Foundation of China NSFC (Grant No. 41874005, 41731066 and 41772365), the Special Project granted by the China Earthquake Administration (Grant No. 2018010103) and the Fundamental Research Funds for the Central Universities, CHD (No. 300102269303, 300102269722).

**Acknowledgments:** The authors would like to thank the European Space Agency for providing the Sentinel-1A SAR data freely. ALOS/PALSAR data were copyrighted by Japan Aerospace Exploration Agency (JAXA) and TerraSAR-X data were provided by DLR. Three-arc-second SRTM DEM is freely downloaded from http://www2.jpl.nasa.gov/srtm/cbanddataproducts.html. And above all, special thanks to three anonymous reviewers of this paper for their constructive comments.

**Conflicts of Interest:** The authors declare no conflict of interest.

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
