# Peer review of "Research on Spatiotemporal Land Deformation (2012–2018) over Xi’an, China, with Multi-Sensor SAR Datasets"

_remotesensing, doi:10.3390/rs11060664_

Round 1

Reviewer 1 Report

General Comments:

1.       A thorough English edit by a native English speaker is required. Many tenses and plural usages are incorrect throughout the manuscript. Additionally, verb, adverb, or adjective choices are sometimes incorrect, leading to confusing sentence structure or misinterpreted intended meanings.

2.       The figure quality throughout the manuscript is excellent. The only exception to this is Figure 2 (the cross section, which is very light and the legend is almost illegible and it’s also unclear why the faults don’t correlate with the plan view map).

3.       Lines 92-95: This is the goal or main purpose of this paper, but this is extremely weak and doesn’t really tell the reader what is unique or what advancements are being made in this paper. It sounds more like a site investigation that is focused on one small area. This part needs to be greatly expanded and you need to be explicit as to how this work advances some technique or the application of the technique provides some revelation of analysis that perhaps couldn’t be previously accomplished. As it is, it’s unclear what advancements are being made, which weakens the paper greatly.

4.       It isn’t clear (not stated) what you deem as sufficient vs. insufficient SAR data, where SBAS supplemental data are needed. Can you be more specific?

5.       The paper is more of a site investigation with descriptions of what is observed with little explanation of how things are occurring. I was disappointed in the discussion of the connection between faults and fissures and overall land subsidence. There was no explanation of the actual hydromechanical processes that led to the observed behavior of the system, merely an explanation of the observation and the connection without specifics of behavior and possible stress scenarios that would lead to the fissures in the first place.

6.       The application of the sill model didn’t really fit the overall flow of this paper. It seemed more of an afterthought and the fact that it was only applied to one small region just didn’t seem to be very meaningful.

7.       I’m still uncertain what advancements are being made here scientifically. The applications and techniques are not new and the results are not unusual, but the discussion lacked substance so that this overall investigation seems more like a site investigation than one in which new science or findings were presented that could be useful in other studies.

Specific Comments:

1.       Title is very clumsy. It implies you’re deriving and analyzing the character of deformation, which is vague because it’s unclear what the character of land deformation means here. Does it mean the way land deformation changes in space or in time (or some other way)? Furthermore, it implies that you’ve already accomplished the characterization part and now you’re deriving it, which makes no sense. Analyzing it makes more sense but overall the title just doesn’t work. I would simply remove the “Charactization of” and this simplifies and clarifies what you’re actually doing.

2.       Line 26: You can not use abbreviations here. These must be spelled out as the reader does not have reference as to their meaning.

3.       Line 29: A flat lying sill model with distributed contractions is extremely vague and provides no framework for the reader to understand what this even means as this is not common word or modeling usage for subsidence investigations

4.       Line 31: Manifests is not the correct word choice here. A better word is “suggests” is better.

5.       Line 45 (and elsewhere): change ‘funnel’ to ‘bowl’ throughout. Funnel is not technically correct unless a single well is responsible for the entire deformation signal. Usually it’s a series of wells and the overall shape is not a funnel but is more bowl shaped, which is the correct expression for localized regions of vertical subsidence.

6.       Line 48: the correct word is “cumulative”

7.       Line 206-207: The method to convert from LOS to vertical here is not adequately described. Simply indicating a method from another paper is not sufficient. At least briefly describe the method used.

8.       Line 272: It’s not obvious that there are 5 subsidence bowls. Only one is truly obvious. The others are more of a zone than a bowl, perhaps associated with the fault. Somehow you need to identify these zones to the reader on the figure because it isn’t clear what the zones are by looking at figure 3.

9.       Line 295: Bowl, not bowels.

10.   Figures 6, 7, and 8: It would be very helpful to include the location of the fissures to all the InSAR plots (b, c, and d) so that the reader can better identify the correlation with ‘a’ relative to the fissures and how they change on an annual basis.

11.   Line 412: You need to change “contraction” to “compaction”, which is the correct term (and elsewhere). This section stresses the importance of including the faults in the cross section of figure 2, because it’s not clear to the reader otherwise what the connection is between the faults and subsidence.

12.   Line 422-423: You indicate that subsidence is influenced by earth fissures. Unless the earth fissures are tectonically induced, this is opposite of reality. In this case, you then need to define what you mean by earth fissures, because in hydrodgeology earth fissures are a result of deformation induced by pumping, not the other way around. Hence, fissures are caused by land subsidence and the mechanisms can be variable (e.g. differential vertical subsidence associated with faults, rotation along the fault, horizontal deformation causing increased stress at the land surface, etc). You fail to adequately address the stress state that may form earth fissures. You merely say the fissures are distributed along the fault but don’t discuss mechanisms, which you need to at least speculate on if you don’t know the answer.

13.   Line 427-428: You need to discuss why subsidence occurs between the fissures. What is happening mechanically here?

14.   Figure 14d: Your axes labels are incorrect. Distance should be on the x-axis and deformation on the y-axis

Author Response

Please see the attached response file.

Reviewer 2 Report

See attached file

Author Response

Please see the attached response file.

Reviewer 3 Report

Dear Authors, you will find my comments in the attached file.

I asked minor revisions for the following two main reasons:

You should better explain how did you convert LOS deformation to vertical deformation. This is a crucial point for your results and their discussion.

In my opinion, you should you have to review and make it clearer Section 4.3.

Best regards

Author Response

Please see the attached response file.

Round 2

Reviewer 1 Report

The authors did a better job of describing the processes of land subsidence and explaining the differences in compaction that are occurring in different areas of the region in the discussion section of the paper.

The authors did address all of my original concerns, but I still contend this is largely a case study and that there’s nothing novel or new being addressed in this paper.

Although the paper describes what the faults do to the overall configuration of the water levels and resulting subsidence. The paper fails to address the hydromechanical properties or conditions of the faults that lead to the resulting configurations. Why are the faults behaving this way? What conditions have occurred that have led the faults to behave as they do? How do the hydromechanical properties of the faults differ from the surrounding aquifer system? These are interesting questions that were not adequately addressed in the paper.

The authors did a much better job of explaining the sill model and its usefulness in this investigation.

The overall language is better but I still found a number of problems that I’ve listed below:

1.       Line 89: Remove the word “been”

2.       Line 93: Change “at” to “for”

3.       Line 94: This sentence makes no sense. The use of “inversed” is not the correct usage here

4.       Table 1: Make the dates more legible by using “/” or “-“ between years, months and days.

5.       Line 402: Change “geology” to “geologic”

6.       Line 403: Since you’ve already indicated geological factors, why not say hydromechanical here?

7.       Line 405: Change “stratus” to “stratigraphic”

8.       Line 413: move “nearby” to in front of the word normal

9.       Line 416: replace current phrasing with “when the water-level declines”

10.   Line 431: Change “distribute” to “occur”, remove the word “area”

11.   Line 438: Should read:  “…act as barriers to groundwater flow”

12.   Line 441: Remove “the” before groundwater

13.   Line 445: Change “in the aquifer” to “to the aquifer”

14.   Line 482: What does “backward conditions” mean? Unclear

15.   Line 524: The heading should be plural “Conclusions” because you have more than one conclusion.

16.   Line 551: “in the future…”. Also, you have put the subject at the end of the sentence. It should really read “In the future it will be necessary to … to better understand…”

Author Response

Please see the attached letter.
